# Effects of Long-Term Sod Culture Management on Soil Fertility, Enzyme Activities, Soil Microorganisms, and Fruit Yield and Quality in “Jiro” Sweet Persimmon Orchard

**DOI:** 10.3390/plants13111573

**Published:** 2024-06-06

**Authors:** Xu Yang, Bangchu Gong, Cuiyu Liu, Yanpeng Wang, Yang Xu

**Affiliations:** Research Institute of Subtropical Forestry, Chinese Academy of Forestry, Hangzhou 311400, China; yangxu2119@126.com (X.Y.); gongbc@126.com (B.G.); ankar_liu@163.com (C.L.); yanpengwang@caf.ac.cn (Y.W.)

**Keywords:** sod culture, “Jiro” sweet persimmon, microbial community structure, fruit yield and quality

## Abstract

Clean tillage frequently causes the loss of soil nutrients and weakens microbial ecosystem service functions. In order to improve orchard soil nutrient cycling, enhance enzyme activities and microbial community structure in a “Jiro” sweet persimmon orchard, sod culture management was carried out to clarify the relationship among soil nutrient, microbial communities, and fruit yield and quality in persimmon orchard. The results showed that sod culture management increased the content of organic matter, total organic carbon, nitrogen, phosphorus, and potassium in the soil, thus improving soil fertility. Compared with clean tillage orchards, sod culture methods significantly increased soil enzyme activities and microbial biomass carbon (MBC) content. The abundance-based coverage estimator (ACE) and the simplest richness estimators (Chao l) indices of the bacterial community and all diversity and richness indices of the fungal community significantly increased in the sod culture orchard, which indicated that sod culture could increase the richness and diversity of the soil microbial community. The dominant bacterial phyla were *Proteobacteria* (32.21~41.13%) and *Acidobacteria* (18.76~23.86%), and the dominant fungal phyla were *Mortierellomycota* (31.11~83.40%) and *Ascomycota* (3.45~60.14%). Sod culture drove the composition of the microbial community to increase the beneficial microbiome. Correlation analyses and partial least squares path modeling (PLS-PM) comparative analyses showed that the soil chemical properties (mainly including soil organic matter content, total organic carbon content, total potassium content, and total nitrogen content), soil enzyme activities and soil microorganisms were strongly correlated with fruit yield and quality. Meanwhile, soil nutrient, soil enzyme, and soil microbes had also influenced each other. Our results showed that long-term ryegrass planting could improve soil fertility, enzyme activities, and microbial community compositions. Such changes might lead to a cascading effect on the fruit yield and quality of “Jiro” sweet persimmons.

## 1. Introduction

The “Jiro” sweet persimmon is one of the main varieties of persimmon that is widely planted in Baoshan City (Yunnan Province, China) [1]. Baoshan City is a dry–hot valley. The area has a deep layer and fertile soil of alluvium [2]. However, there are still many problems associated with sweet persimmon planting in this area. An investigation of “Jiro” persimmon orchards revealed that most of them use traditional farming methods that can cause imbalance in orchard ecology. The soil organic matter content was lower than the theoretical values. There was a lower available soil P and Fe content in 47.37% of the orchards and a higher N content in 84.21% of the orchards. Therefore, the amount of organic matter and P and K fertilizer should be increased, and the amount of N fertilizer should be decreased [3].

Sod culture is a soil management practice in orchards, in which grass is planted to cover trees or whole orchards. This management is common in European, North American, and Japanese orchards, which constitute 55~90% of the total orchard area in these countries [4]. In contrast, less than 20% of the orchard area in China implements these measures because of the false understanding that grass will compete for fertilizer with fruit trees [5]. First, orchards always involve the management of clean tillage. The increased tillage intensity accelerated soil degradation, reduced the mechanical stability and content of soil aggregate [6], and increased soil viscosity and hardening (with the 50.91% of soil clay) [7], thus causing severe soil secondary salinization [8]. Second, insufficient organic fertilizer sources and the highly effective biodegradation of organic matter continuously decrease orchard soil quality, affecting the nutrient uptake and utilization of plants [9,10]. Finally, traditional farming practices also caused a rapid decrease in soil microbial biomass, which causes a reduction in microbial quotient [11]. Sod culture was an effective way to overcome sheet erosion, accelerate the formation of soil aggregate structures, decrease bulk density [12], increase total porosity, and improve the structural characteristics of the soil [13]. Additionally, sod culture could restrain soil leaching and increase the content of organic and mineral elements and the activity of beneficial soil microbes [14,15]. Sod culture has achieved satisfactory ecological and economic benefits in apple [13], orange [16], pear [17], and other fruit orchards.

There are few studies on sod culture in persimmon orchards. Previous studies showed that sod culture was useful for reducing the occurrence of photosynthetic noon breaks in the “Mopan persimmon” [18] and increasing the soil microbial population and enzyme activity in “Youhou” sweet persimmons [19]. Moreover, sod culture also influenced soil nutrients and fruit quality, such as increased soil fertility, improved soil physicochemical properties, and increased fruit yield, vitamin C content, and total sugar content in the “shuishi persimmon” [20]. However, co-occurring patterns exist among microbiological, plant, and environmental factors [21]. The plant–soil–microorganism system influences the nutrient cycle and energy flow of agroecological systems. In this system, microorganisms play an important role. However, the most important changes after the sod culture of persimmon orchard soil and the interactions among the plant–soil–microorganism system are still unknown.

In this study, we compared the differences in soil fertility, enzyme activities, and soil microorganism communities in clean tillage and sod culture orchards. The synergistic changes in “Jiro” sweet persimmon fruit yield and quality were also analyzed. In addition, the relationship among the plant–soil–microorganism system was further discussed. This study aimed to answer the following questions: (1) whether the difference in soil chemical properties and microbiological compositions are induced in persimmon orchards by the sod culture method; (2) how this method influences the changes in soil nutrients, enzyme activities, and soil microbes during the course of persimmon development; and (3) what the relationship is among soil, microbes, persimmon fruit yield, and the interactions among these factors. This study may help to provide a theoretical foundation for cultivation and management, habitat adjustment, and soil condition evaluation in “Jiro” sweet persimmon orchards.

## 2. Results

### 2.1. Impacts on Soil Chemical Properties

There were significant differences in the soil chemical properties among the different treatments (Table 1). Total organic carbon (TOC), total nitrogen (TN), hydrolyzed nitrogen (HN), and available phosphorus (AP) content in treatments A and C were significantly greater than those in treatments B and D, which indicated that these indices improved with increasing the herbage planting period. Soil organism (SOM), TN, total phosphorus (TP), TOC, HN, available potassium (AK), and AP contents significantly increased in the sod culture soil of the 5-year-old (treatment A) and 8-year-old (treatment C) persimmon orchards compared with those in the clean tillage soil (treatments B and D). Compared with those in treatment B, 22.14%, 54.33%, 70.31%, 36.22%, 26.46%, 107.79%, and 81.98% of the indices increased in treatment A, and compared with those in treatment D, 13.91%, 12.43%, 21.59%,59.20%, 17.02%, 144.76%, and 82.97% of the indices increased in treatment C.

### 2.2. Impacts on Soil Enzyme Activity

Soil enzyme activities were increased with the increasing herbage planting period both in sod culture and clean tillage orchards (Table 2). There were significant differences in the soil enzyme activity between sod culture soil and the contrast except soil catalase. Compared with those in treatment B, 39.65%, 8.95%, 25.86%, and 40.50% of the urease, catalase, catalase, and alkaline phosphatase activities increased in treatment A, and compared with those in treatment D, 34.29%, 3.43%, 24.39%, and 41.30% of the indices increased in treatment C.

### 2.3. Impacts on Soil Microbial Biomass Carbon and Microbial Diversity

As a characteristic index of soil active carbon, soil microbial biomass carbon (MBC) can reflect soil nutrient cycling, translocation, and transport. There were significant differences among treatments in the persimmon orchards (Figure 1). The MBC content increased with increasing herbage planting period. Meanwhile, the MBC contents increased 32.33% and 73.88%, respectively, in the sod culture soil of the 5-year-old (treatment A) and 8-year-old (treatment C) persimmon orchards, compared with those in the clean tillage soil (treatments B and D), which indicated that sod culture is helpful for improving the quantity and activity of soil microbes.

The amplicon sequencing results revealed 72,510 effective bacterial sequences and 6626 effective fungal sequences. The numbers of bacterial OTUs and fungal OTUs in the four treatments were 3253.00~3526.00 and 48.33~119.67, respectively (Table 3). There was no significant difference in the OUTs between treatments. The OUTs of treatments A and C were slightly greater than those of treatments B and D, which indicated that the sod culture may have had little impact on the soil bacterial or fungal community number.

The α-diversity indices of the bacterial community in treatments C and D were greater than those in treatments A and B, which means that the duration of life increased the bacterial diversity. Similarly, the ACE, Chao1, and Shannon indices in treatments A and C were higher than those in treatments B and D, which indicated that sod culture could enhance bacterial community diversity. However, there was no significant difference in the diversity indices among the four treatments. There were significant differences in all fungal community diversity indices among the orchards with different planting durations, as well as among the sod culture and control orchards. These findings indicated that sod culture enhanced the diversity and richness of the fungal community (Table 3).

### 2.4. Impact on the Soil Microbial Community Composition

Based on an overall evaluation of the soil bacterial communities in the different treatments, the five most abundant phyla were *Proteobacteria*, *Acidobacteria*, *Bacteroidota*, *Gemmatimonadota*, and *Actinobacteriota*, accounting for 74.01–79.57% of the total bacterial abundance in the persimmon orchard, among which *Proteobacteria* and *Acidobacteria* were the most abundant bacterial phyla in the soil, accounting for 32.21~41.13% and 18.76~23.86%, respectively (Figure 2). The amount of *Proteobacteria* increased (from 37.13% in treatment B to 44.01% in treatment D) over time. Sod culture significantly increased the abundance of *Proteobacteria* (from 32.21% in treatment B to 37.13% in treatment A and from 41.43% in treatment D to 44.01% in treatment C) and slightly decreased the abundance of *Acidobacteria* (from 23.86% in treatment B to 21.69% in treatment A and from 20.15% in treatment D to 18.77% in treatment C). At the bacterial genus level, the abundance of all genera changed slightly after sod culture, which means that the bacterial communities were not sensitive to the changed environment.

At the fungal phylum level, *Mortierella* and *Ascomycota* were abundant compared to the other fungi and accounted for 31.11~83.40% and 3.45~60.14%, respectively, of the fungal community. Similarly, with prolonged growth time. the abundance of *Ascomycota* significantly increased (from 3.45% in treatment B to 60.14% in treatment D). After sod culture, the number of *Ascomycota* decreased, and the number of *Mortierella* significantly increased. At the fungal genus level, *Mortierella* and *Chaetomium* were the dominant genera in treatments, with proportions ranging from 31.48 to ~83.39% and 0.01 to ~55.83%, respectively. However, *Mortierella* increased most significantly, and *Chaetomium* slightly decreased after sod culture. In addition, the numbers of *Plectosphaerella*, *Saitozyma*, and *Fusarium* increased.

The PCoA also showed that there was little difference between and within groups of the sod culture and control orchard on bacterial communities (Figure 3a). However, the species composition was significantly different between groups but similar within groups, which indicated that the sod culture largely changed the fungal community (Figure 3b).

### 2.5. Correlation between Soil Chemical Properties and Microbial Community Structure

A Spearman analysis was conducted to study the relationships between soil chemical properties and microbial community structure (Figure 4). At the genus level, the top ten bacterial (Figure 4a) and fungal (Figure 4b) genera had significant or inconspicuous negative or positive correlation with soil pH, TP, TN, TOC, and AN content, while among the main bacterial genera, *Lysobacter* also had negative correlation to the AP content.

### 2.6. Impacts on Persimmon Yield and Fruit Quality

Compared with those in clean tillage orchards, the sod culture method increased the fruit transverse length (FTL), fruit weight (FW), single-plant yield (SPY), and soluble solids content (SSC) of “Jiro” sweet persimmons. However, this method had little influence on fruit firmness (FF). The FW, SSC, and SPY of treatments A and C increased by 8.89~9.32%, 4.52~5.67%, and 10.41~14.14% compared with those of treatments B and D, respectively (Table 4).

### 2.7. Plant–Soil–Microbe Interactions

From the soil fungal–bacteria interaction network (Figure 5), we found that *Pseudomonas*, unclassified *Alphaproteobacteria*, and unclassified *Rokubacteriales* were the top three bacterial taxa, while *Pseudaleuria*, *Malassezia*, and *Ramophialophora* were the top three fungal taxa that contributed to fruit weight. Unclassified *Vicinamibacteraceae*, unclassified *Gemmatimonadaceae* and unclassified *SC_I_84* were the top three bacterial taxa, while *Chaetomium*, *Plectosphaerella*, and *Fusarium* were the top three fungal taxa in contributing to the single-plant yield and soluble solids content.

The partial least squares path modeling (PLS-PM) of soil chemical properties, soil enzyme activities, soil microorganism, and persimmon fruit yield and quality in different treatments were conducted in this study. The results showed that soil chemical properties (mainly including TN, TP, TOC, and SOM), soil enzyme activity, and soil microorganism were significantly positively related to fruit yield and quality. Soil chemical properties had more influence on fruit yield and quality than soil enzyme activity and soil microorganisms. Soil chemical properties, soil enzyme activity, and soil microorganisms were linked into mutually influential communities and networks (Figure 6).

## 3. Discussion

### 3.1. Impacts of Sod Culture on Soil Physicochemical Properties

With prolonged growth time in monoculture orchards, the decrease in litter chemical diversity weakens the synergistic effects of mixed litter decomposition, thus directly hindering overall decomposition and gradually decreasing soil nutrients [22,23]. On the one hand, herbage growth significantly reduces soil bulk density, increases soil porosity, soil moisture content, and drainage capacity, and increases the nutrient deliverability of soil aggregates, thus increasing organic and mineral nutrient contents [12,13,24]. On the other hand, the increase in aboveground litter and root biomass ensures a reliable, stable source of soil nutrients and mineral nutrition [25]. The increased litter substrate quality stimulated the activity of litter lignocellulose to accelerate litter decomposition [26], thus enhancing the conversion and storage rate of soil nutrients. Therefore, compared with clean tillage, sod culture, especially for long chronosequences, is an ecologically beneficial practice for the recovery of degraded soils. With the increase in the herbage planting period, the soil chemical properties of orchards have improved [27]. The effects of sod cultivation on soil nutrient contents greatly depend on environmental and grass properties. The potential properties of plantgrass, such as biomass, N fixation capacity, litter quality, and root exudates, determine the quality and quantity of substrate inputs to soil food webs and, thus, influence soil nutrient conditions [28]. Ryegrass is a kind of Gramineae species that can significantly change the physical and chemical properties, microbiota, and secondary metabolites of rhizosphere soils [29] and had a good performance on soil chemical properties in our study.

Furthermore, soil physicochemical properties and microbial communities are linked to communities and networks with mutual influence. Soil bacteria and fungi play important roles in pedological mass circulation, soil organic matter decomposition, and the maintenance of soil nutrient balance [30,31]. The mechanism of action of key soil environmental factors changes due to the abundance of soil nutrients after sod culture, thus increasing the ease of microflora substrate usage, significantly affecting the relative abundance of the dominant soil microbial community and increasing the abundance of beneficial microorganisms [32]. These are the main key environmental factors impacting microbial community succession and count promotion. For example, the soil properties of N, P, and TOC directly affected bacterial composition and potential functions [33]. Functional associations were found between key soil properties and oil arbuscular mycorrhizal fungi, and the community was mainly affected by the SOM, AP, and TN contents [34]. On the other hand, the microbial community structure determines the rate of nutrient cycling, organic matter decay, and energy flow, thus promoting the cycling of C, N, and K in soil and increasing soil nutrient content. The microbial community structure determines the rate of nutrient cycling, organic matter decay, and energy flow. For example, *Pseudolabrys* could fix soil nitrogen [35], and the relative abundance of *Blastocladiomycota* might be significantly positively correlated with the soil organic carbon content [36]. Soil bacterial could use simple organic matter to promote photoheterotrophic growth, thus promoting the cycling of C, N, S, and other elements in soil and increasing the nutrient content in soil [37]. In our study, sod culture increased soil SOM and mineral contents, as well as the number of OTUs and community diversity of microorganisms, which indicated that sod culture provided better living conditions for enhancing microbial diversity by improving the soil physicochemical properties.

### 3.2. Impacts of Sod Culture on Soil Enzyme Activity

Soil enzymes can participate in nutrient cycling and transformation extensively and are key properties to reflect soil fertility [38]. This study indicated that compared with clean tillage, sod culture enhanced the activity of soil urease, catalase, sucrase, and phosphatase significantly. The result was same as [39,40]. This was probably because increased plant survivor and root exudates produced by sod culture contained large amount of cellulose, water-soluble polysaccharide, and protein. A sufficient carbon source provided sufficient nutrition and had a direct influence on the growth and propagation of microorganisms. This is helpful for increasing soil microbial activity and enzymatic activity. Meanwhile, more plant survivors were broken down and added into the soil, resulting in an increase in enzyme substrates and a possible enhancement of soil enzyme activities [41].

### 3.3. Impacts of Sod Culture on Soil Microbial Diversity and Population Structure

In addition to maintaining soil microbial richness and diversity indies, sod culture also increases the content of soil microbial biomass carbon, impacts the microbial community composition, increases the community stability of microorganisms, and improves the development of effective microbial communities. With prolonged growth time in tillage soil, the abundance of *Proteobacteria* significantly increased. *Proteobacteria* in the “Jiro” persimmon orchard soil was mainly concentrated in the *Lysobacter*, *Sphingomonas*, *Pseudomona*, and *Allorhizobium* genera. *Lysobacter* is a functional genus related to stress resistance, and other bacterial genera are common orchard pathogens that can cause problems such as *bacterial stem wilt*, leaf blight, and grapevine crown gall disease [42,43,44,45]. An increase in the abundance of *Proteobacteria* may be correlated with the soil stress caused by an increase in the abundance of pathogenic bacteria and a decrease in the abundance of beneficial bacteria.

In this study, the most abundant bacteria were *Proteobacteria* and *Acidobacteria*. This was mainly because these two phyla had greater ecological niches and greater environmental suitability [46]. In our main bacterial community, *Proteobacteria*, *Bacteroidetes*, *Actinobacteria*, and *Nitrospirae* were copiotrophic bacteria, while *Acidobacteria* and *Chloroflexi* were oligotrophic bacteria [47]. The copiotrophic bacteria significantly increased (from 43.92% in treatment B to 48.46% in treatment A and from 54.15% in treatment D to 57.64% in treatment C), while the oligotrophic bacteria significantly decreased (from 30.45% in treatment B to 27.18% in treatment A and from 24.44% in treatment D to 22.11% in treatment C) after sod culture. The soil bacterial communities changed from the slow growth type to the fast growth type. An increase in water conservation and nutrient maintenance ability after sod culture steadily increased the copiotrophic bacteria because of their nutritional preference. Smit proposed the hypothesis that the ratio (P/A) between *Proteobacteria* (P) and *Acidobacteria* (A) may reflect soil nutrient levels [48]. Specifically, a high value indicates sufficient soil nutrition, whereas a low value indicates the opposite. The P/A ratio increased from 1.34 in treatment B to 1.72 in treatment A and from 2.06 in treatment D to 2.35 in treatment C, indicating that the soil nutrient level in the sod culture orchard was greater than that in the tillage orchard.

Compared with that of soil bacteria, the fungal community composition changes dramatically. Under control conditions, the speed of bacterial growth and turnover is rapid. Therefore, bacteria are likely to have greater resistance and resilience to external factors, and their sensitivity to changes in living conditions is lower than that of the fungal community [49]. With prolonged growth time in clean tillage soil, the amount of *Ascomycota* significantly increased. The *Ascomycota* in the “Jiro” persimmon orchard soil was mainly associated with pathogenic fungi, such as *Verticillium* [50] and *Thelonectria* [51], and with antibiological inoculants, such as *Penicillium* and *Trichoderma* [52]. An increase in *Ascomycota* indicated unhealthy living conditions in the soil fungal community. *Mortierella* spp. are the main saprophytic trophic fungi that acquire nutrients and energy by resolving plant residue, which can degrade cellulose, hemicellulose, and lignin [53]. An increase in litter residue caused this fungus to have a strong competitive effect on soil fungi after sod culture. *Ascomycota* grow well in fertile soil [54]. *Chaetomium* can inhibit plant pathogenic fungi and has an antagonistic effect on many fungi, such as *Colletotrichum gloeosporioides*, *Cladosporium fulvum*, and *Bipolaris sorokiniana*. An increase in *Chaetomium* indicated the improvement of fungal communities.

### 3.4. Impacts on Sod Culture on Fruit Yield and Quality

Fruit yield and quality are the main factors for the benefits of persimmon orchards. After improving soil structure and nutrition and changing soil microflora, plant growth such as root growth, leaf nutrient content, and photosynthetic efficiency is impacted, thus impacting the output and quality of the fruit [55]. Sod culture can provide a nitrogen pool, control nitrogen loss, and provide N for plant-appropriate growing [28]. Furthermore, sod culture also enhanced the soil SOM, TN, and NH_4_^+^-N content, promoted leaf photosynthetic capacity, and enhanced leaf N content and fruit sugar metabolism [56], thus increasing fruit-soluble sugar, soluble solids, and vitamin C content [12]. On the other hand, sod culture can influence fruit yield and quality by changing microflora composition, improving the abundance of soil bacteria and fungi, and increasing microbial enzymatic activity. Cover crops could improve microbial biomass and alter beneficial microbial functions by strengthening the links between microorganisms in orchards, thus impacting apple and strawberry growth, fruit production, and nutritional quality [40,57]. In our study, soil chemical properties (such as SOM, TN, TP, and TOC) and bacterial and fungal diversity were significantly positively correlated with fruit yield and quality (FW, SPY, and SSC), indicating that changes in soil chemical properties and soil microorganisms caused by sod culture were beneficial for improving persimmon fruit yield and quality.

## 4. Materials and Methods

### 4.1. General Situation of the Experimental Place

The experiment was conducted at Dianhong shuoguo sweet persimmon plantations (E99°12′39.55″, N23°11′53.06″, altitude 1653.5 m) in Banqiao Town, Longyang County, Baoshan, Yunnan, China. This area has a subtropical monsoon climate, with an annual average temperature of 15.5 °C, annual precipitation of 966.5 mm, and an annual frost-free period of more than 290 days. The experimental soil is paddy soil with moderate fertility.

### 4.2. Test Material

The experiments were carried out in 2015 and 2018 to explore the effects of long-term sod culture on persimmon orchards. The rootstock–scion combinations used were *Diospyros lotus* and the “Jiro” sweet persimmon, with line and row spacings of 4 m × 3 m. We planted *Lolium perenne* at the beginning of orchard establishment (Treatment A and C), and each plot was spaced 1 m apart and extended 150 m in length and 2 m in width. We cut the grass 3~4 times each year and covered it under the trees. The contrasts were derived from clean tillage soil of the same age and cultivation methods in the plantation (Treatment B and D). All experiments were conducted using a randomized complete block design with three replications.

### 4.3. Soil Sample Collection

The 0~30 cm soil layer samples were collected from 30 to ~40 cm from the trunk at five locations according to the “S” sampling method on 20 September 2023. Approximately 500 g of soil sample from each plot was divided into three groups after impurities were removed and fully mixed as follows: 5 g of the sample was packed into sterile tubes and placed in a −80 °C freezer for microorganism analysis; a 20 g fresh soil sample was taken for the determination of microbial biomass carbon (MBV) content; and the other sample was sieved through a 0.15 mm sieve after air drying for the determination of soil chemical properties and soil enzyme activities.

### 4.4. Test Methods

#### 4.4.1. Determination of Soil Chemical Indices

The soil pH value was measured using the potentiometric method (decarbonized water to soil ratio of 2.5:1, Mettler Toledo Seven-Excellence S470, Zurich, Switzerland). The soil organisms (SOM) were tested using the potassium dichromate method. The total nitrogen (TN) content was tested using the Kjeldahl determination. The hydrolyzed nitrogen (HN) content was tested using the alkalotic diffusion method. The total potassium (TK) and available potassium (AK) contents were tested using flame photometry. The total phosphorus (TP) and available phosphorus (AP) contents were tested using molybdenum antimony resistance colorimetry. The soil total organic carbon (TOC) content was tested using the dichromate titration method with Toc-ssm-5000A (Shimadzu, Japan) [58]. All experiments were repeated three times.

#### 4.4.2. Determination of Soil Enzyme Activities

The soil urease (S-URE), soil catalase (S-CAT), soil sucrase (S-SUE), and soil alkaline phosphatase (S-AKP) activities were determined using an S-URE, S-CAT, S-SUE, and S-AKP enzymatic kit (Suzhou Comin Biotechnology Co. Ltd., Suzhou, China) with ELIASA (Molecular Devices, Sunnyvale, CA, USA) at 578, 240, 510, and 660 nm wavelengths, respectively. Soil urease activity, expressed as milligrams of NH_3_-N, was produced in 1 g of soil after 1 d of incubation; catalase activity, expressed as milliliters of 0.1 mol L^−1^ KMnO_4_, was consumed in 1 g h^−1^ of soil; sucrase activity, expressed as milligrams of glucose, was produced in 1 g of soil after 1 d of incubation; and alkaline phosphatase activity, expressed as milligrams of phenol, was released from 1 g of soil after 1 d. Each soil sample was measured 3 times, and the average was taken [59].

#### 4.4.3. Determination of Soil Microbial Biomass Carbon Content

Soil microbial biomass carbon (BMC) content was tested using the chloroform fumigation and leaching method. Chloroform fumigated and unfumigated soils were extracted with 0.5 mol L^−1^ K_2_SO_4_ solutions at a soil/liquid ratio of 1:4. Microbial biomass extracts were analyzed for organic C on a Shimadzu TOC-L and TNM-L analyzer (Shimadzu Corporation, Kyoto, Japan). The average of three replicate measurements of fumigation and non-fumigation conditions was calculated [60].

#### 4.4.4. Soil DNA Extraction, Amplification, and Sequencing

Total DNA was extracted from the persimmon soil microbes using a Fast DNA Spin kit (MP Biomedicals, Santa Ana, CA, USA). The purity and integrity of the DNA were checked using a NanoDrop 2000 spectrophotometer (Thermo Scientific, Wilmington, DE, USA) and 1% agarose gel electrophoresis. The above extracted DNA was used as a template, after which specific primers were used to amplify the bacterial 16S rRNA V3 + V4 region with barcodes 338-F (5′-ACTCCTACGGGAGGCAGCA-3′), 806-R (5′-GGACTACHVGGGTWTCTAAT-3′), ITS1-F(5′-CTTGGTCATTTAGAGGAAGTAA-3′), and ITS4-R (5′-TCCTCCGCTTATTGATATGC-3′), which were used to amplify the fungal ITS1 region. The PCR procedure is described as follows: 98 °C for 2 min, 30 cycles; 98 °C for 30 s, 50 °C for 30 s, and 72 °C for 30 s; and 72 °C for 5 min. The PCR products were examined using 2% agarose gel electrophoresis. A DNA purification kit (OmegaBiotek, Norcross, GA, USA) was used to create the library. After the library was constructed and qualified through quantification and library testing, sequencing was performed on the Illumina NovaSeq6000 PE250 platform [61,62].

#### 4.4.5. Fruit Yield and Quality Detected

Persimmon fruits were harvested to estimate the yield and the quality of each treatment. We checked the number of fruits per plant of each treatment in September 2023. Fifty fruits per treatment were randomly selected from the north, south, east, and west directions of the canopy. The fruit longitudinal length (FLL) and fruit transverse length (FTL) were measured with a vernier caliper, and fruit weight (FW) was measured with electronic scales. Then single-plant yield (SPY) of each treatment was calculated with the fruit number and mean FW. Fruit firmness (FF) was measured with a fruit hardness tester (GY-4, Zhejiang TOP Cloud-agri Technology, Hangzhou, China), and the soluble solids content (SSC) was measured with a saccharometer (TD-45, Zhejiang TOP Cloud-agri Technology, Hangzhou, China). All experiments were repeated three times.

### 4.5. Data Analysis

OTUs were obtained with Usearch software (version 10.0) and clustered at a 97% similarity level. Alpha diversity estimators were performed with Mothur (version 1.30). Principal coordinate analysis (PCoA) was performed with the vegan toolkit in the R language tool. The analyses of the relationships between microorganisms and soil environment factors were performed via redundancy analysis (RDA) in R (V2.3). A two-factor correlation network analysis of the root–soil–microbe interactions was performed using the NetworkX toolkit in R. The contributions of the soil environmental factors to persimmon yield and quality were analyzed via a structural equation model in SmartPLS 4 software, one-way analysis of variance (*p* < 0.05) was analyzed with SPSS 20.0 software, and the tables were created using Excel 2010.

## 5. Conclusions

Based on the results, soil nutrients, soil enzymes, and soil microorganisms were linked into mutually influential communities and networks. Sod culture in persimmon orchards obviously increased the soil nutrient content and enzyme activities, significantly enhanced the soil microbe biomass C content and the diversity of the soil microflora, drove the composition of the microbial community to increase the beneficial microbiome, and greatly increased that of the eutrophic type, thus improving persimmon fruit yield and quality. This information further increased our understanding of nutrient cycling and transformation in terrestrial ecosystems of the soil–microorganism–plant system. The farmers’ incomes increased through the increase in “Jiro” sweet persimmon fruit yield and quality, which made orchard grass mulching management easy to generalize in the Baoshan area. In order to choose more suitable culture varieties to adapt to local climate and soil conditions, additional studies needed to be undertaken in future work.

## Figures and Tables

**Figure 1 plants-13-01573-f001:**
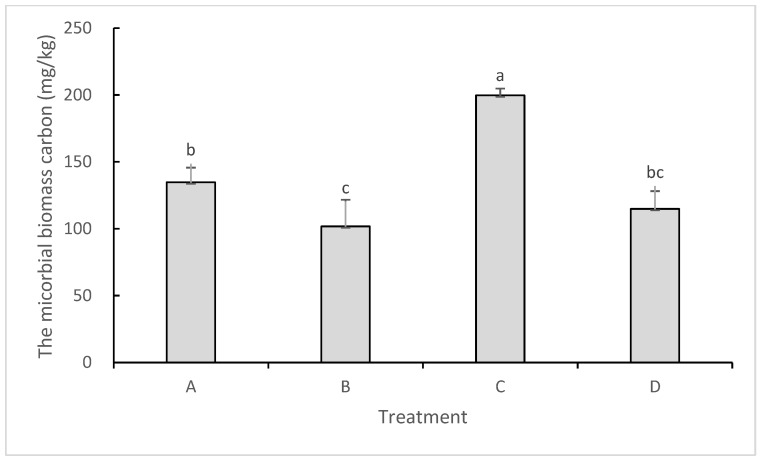
Soil microbial biomass carbon content (different small letters indicate the significance of the difference).

**Figure 2 plants-13-01573-f002:**
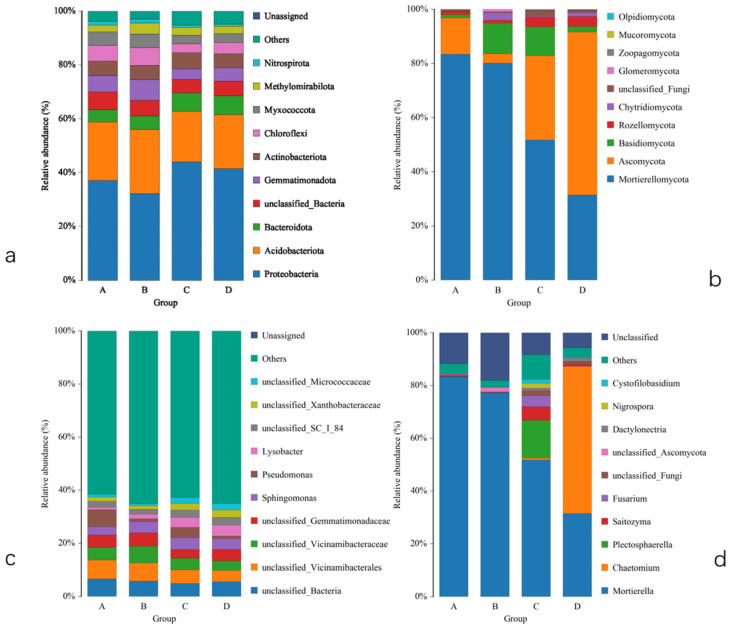
Bacterial and fungi community composition and structure from phyla (**a**,**b**) to genera (**c**,**d**) of the relative abundance. They are plotted by the “Relative Abundance” on the *Y*-axis and “Group Name” on the *X*-axis.

**Figure 3 plants-13-01573-f003:**
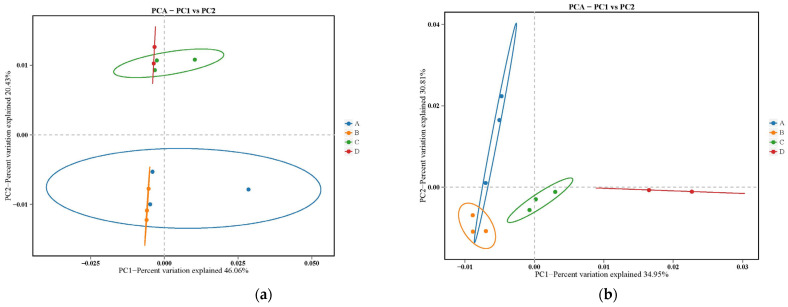
Cluster analysis of PCoA community structure of soil bacteria (**a**) and fungi (**b**).

**Figure 4 plants-13-01573-f004:**
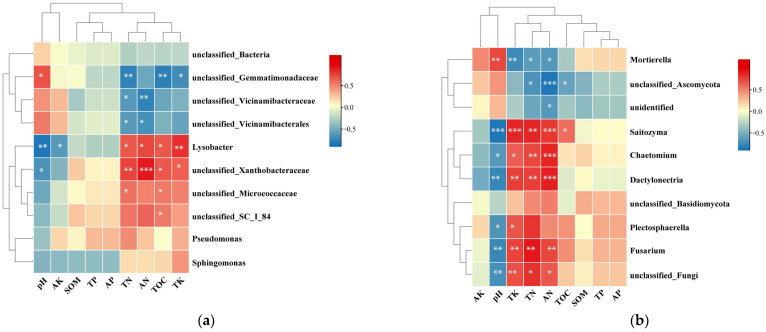
Spearman correlational analysis of environmental factors and soil bacteria (**a**) and fungi (**b**) communities (genus level). Statistically significant differences are indicated (* *p* < 0.05; ** *p* < 0.01; *** *p* < 0.001).

**Figure 5 plants-13-01573-f005:**
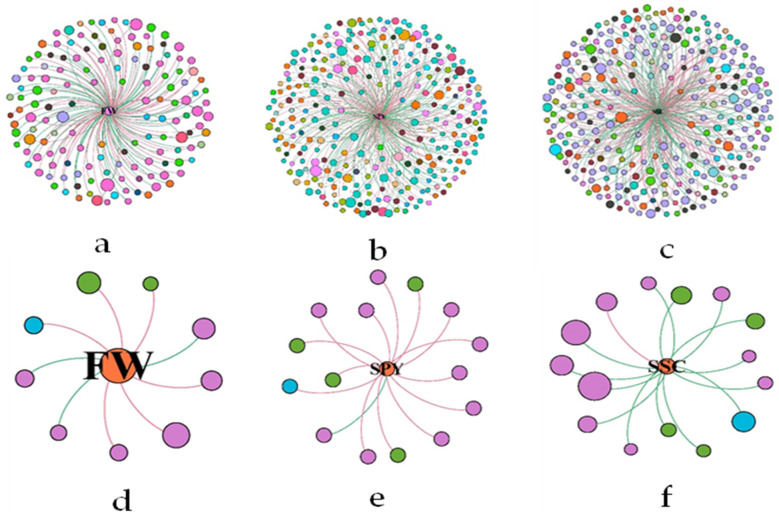
Network analysis of two-factor correlations for fruit yield and quality indices and major bacterial and fungal taxa genera. Taxon–taxon networks of bacteria (**a**–**c**) and fungi (**d**–**f**) in the three fruit yield and quality indices (FW, SPY, and SSC). In the panels, the connections indicate strong and significant (*p* < 0.05) correlations; nodes represent unique sequences in the datasets; and the same color indicates one modularity. For detailed network properties, see Appendix A.

**Figure 6 plants-13-01573-f006:**
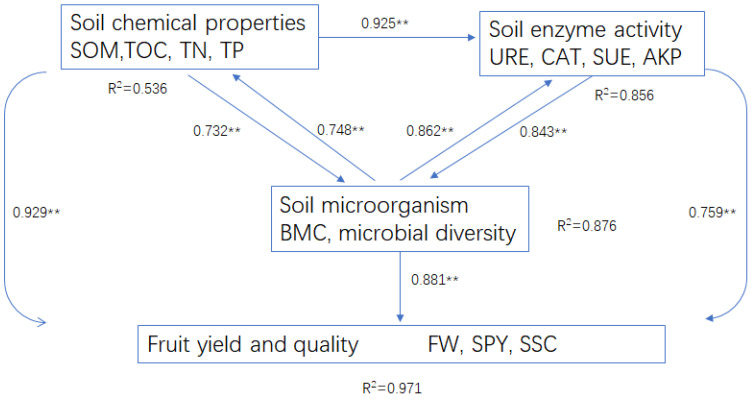
Partial least squares path modeling (PLS-PM) of main soil chemical properties (SOM, TOC, TN, and TP), soil enzyme activities, soil microorganisms, and fruit yield and quality. Path coefficients (i.e., direct effects) are displayed on arrows, and ** indicates that the path coefficient is significant (*p* < 0.01). The R^2^ values represent the variance of the dependent variables explained by the inner modal.

**Table 1 plants-13-01573-t001:** Chemical properties of the different persimmon orchard soils (Mean ± S.D.).

**Treatment**	**pH Value**	**TN (g/kg)**		**TK (g/kg)**		**TP (g/kg)**		**TOC (g/kg)**
A	7.48 ± 0.91 a	1.93 ± 0.18 a		9.51 ± 0.13 c		1.09 ± 0.16 a		11.17 ± 041 c
B	7.46 ± 0.35 a	1.29 ± 0.19 b		10.42 ± 0.22 b		0.64 ± 0.15 b		8.20 ± 0.25 d
C	7.41 ± 0.40 a	1.99 ± 0.16 a		18.11 ± 0.16 a		1.09 ± 0.13 a		27.70 ± 1.05 a
D	7.42 ± 0.03 a	1.77 ± 0.14 a		18.03 ± 0.17 a		0.88 ± 0.14 ab		17.40 ± 0.17 b
**Treatment**	**HN (mg/kg)**		**AK (mg/kg)**		**AP (mg/kg)**		**SOM (g/kg)**	
A	119.14 ± 2.05 c		201.95 ± 2.24 a		96.02 ± 0.25 b		32.04 ± 1.89 a	
B	93.94 ± 0.41 d		111.17 ± 1.71 c		46.20 ± 0.18 d		26.19 ± 0.16 b	
C	154.87 ± 2.38 a		143.97 ± 3.41 b		126.02 ± 2.14 a		31.09 ± 0.86 a	
D	131.01 ± 0.44 b		78.73 ± 0.18 d		51.63 ± 0.23 c		27.30 ± 0.08 b	

Note: Total nitrogen (TN), total potassium (TK), total phosphorus (TP), total organic carbon (TOC), hydrolyzed nitrogen (HN), available potassium (AK), available phosphorus (AP), soil organism (SOM). Different lowercase letters in the figure indicate significant differences (*p* < 0.05 by LSD test) among treatments.

**Table 2 plants-13-01573-t002:** Soil enzyme activity of the different persimmon orchard soils (Mean ± S.D.).

Treatment	Urease (mg g^−1^ d^−1^)	Catalase (mL g^−1^ d^−1^)	Sucrase (mg g^−1^ d^−1^)	Phosphatase (mg g^−1^ d^−1^)
A	0.61 ± 0.03 b	0.34 ± 0.03 ab	2.19 ± 0.08 c	6.73 ± 0.14 c
B	0.44 ± 0.01 d	0.31 ± 0.01 b	1.74 ± 0.02 d	4.79 ± 0.19 d
C	0.67 ± 0.03 a	0.37 ± 0.03 a	2.55 ± 0.08 a	13.01 ± 0.41 a
D	0.50 ± 0.10 c	0.36 ± 0.02 a	2.05 ± 0.03 b	9.21 ± 0.30 b

Note: Different lowercase letters in the figure indicate significant differences (*p* < 0.05 by LSD test) among treatments.

**Table 3 plants-13-01573-t003:** Microbial diversity of persimmon orchard soil.

	Treatment	OTUs	ACE Index	Chao l Index	Shannon Index	Simpson Index
Bacterial community	A	3457.67 ± 133.77 a	3472.37 ± 133.10 a	3292.51 ± 256.85 a	3292.56 ± 259.42 a	0.99 ± 0.00 a
	B	3253.00 ± 214.89 a	3265.20 ± 214.19 a	3253.97 ± 214.33 a	3253.98 ± 214.33 a	0.96 ± 0.00 a
	C	3600.07 ± 295.47 a	3612.29 ± 290.54 a	3568.96 ± 496.73 a	3335.01 ± 491.11 a	1.00 ± 0.00 a
	D	3526.00 ± 60.02 a	3576.01 ± 57.68 a	3236.06 ± 57.58 a	3268.28 ± 280.38 a	1.00 ± 0.00 a
Fungal community	A	63.67 ± 17.78 b	82.84 ± 8.26 c	73.44 ± 12.16 c	0.67 ± 0.07 bc	2.22 ± 0.35 bc
	B	48.33 ± 2.52 b	70.21 ± 12.67 d	59.75 ± 7.33 d	0.58 ± 0.04 c	1.92 ± 0.21 c
	C	119.67 ± 8.51 a	130.40 ± 18.37 a	129.37 ± 7.98 a	0.87 ± 0.04 a	3.99 ± 0.45 a
	D	113.00 ± 20.66 a	106.73 ± 9.73 b	106.10 ± 22.14 b	0.79 ± 0.10 ab	2.96 ± 0.59 b

Note: Operational taxonomic units (OTUs), alpha diversity index of microbial community richness (ACE and Chao 1 indexes) and diversity (Simpson and Shannon diversity indexes). Different lowercase letters in the figure indicate significant differences (*p* < 0.05 by LSD test) among treatments.

**Table 4 plants-13-01573-t004:** Persimmon fruit yield and quality in response to different treatments.

Treatment	FW (g)	FTL (mm)	FLL (mm)	SPY (kg)	FF (kg.cm^−2^)	SSC (%)
A	249.82 ± 29.111 a	54.89 ± 2.91 ab	81.11 ± 5.04 a	44.31 ± 3.73 c	7.04 ± 0.71 a	14.56 ± 0.87 a
B	228.52 ± 25.13 bc	54.51 ± 2.37 b	79.76 ± 4.16 a	38.82 ± 2.75 d	7.09 ± 0.49 a	13.93 ± 0.93 bc
C	239.11 ± 30.60 ab	56.06 ± 2.25 a	81.37 ± 4.63 a	62.01 ± 3.40 a	7.07 ± 0.90 a	14.33 ± 1.52 ab
D	219.59 ± 30.53 c	53.79 ± 3.15 c	79.90 ± 3.29 a	56.16 ± 3.20 b	6.85 ± 0.58 a	13.56 ± 1.12 c

Note: Fruit weight (FW), fruit transverse length (FTL), fruit longitudinal length (FLL), single-plant yield (SPY), FF (Fruit firmness), soluble solids content (SSC). Different lowercase letters in the figure indicate significant differences (*p* < 0.05 by LSD test) among treatments.

## Data Availability

Data are contained within the article and Appendix A.

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
