# Peer review of "Effects of Long-Term Sod Culture Management on Soil Fertility, Enzyme Activities, Soil Microorganisms, and Fruit Yield and Quality in “Jiro” Sweet Persimmon Orchard"

_plants, 2024, doi:10.3390/plants13111573_

Round 1
Reviewer 1 Report
Comments and Suggestions for Authors
Authors have well presented the problem and justified the work. I have some minor points/concerns regarding the methodology. It would be great if authors can analyze the: a) soil enzyme assays, b) microbial biomass carbon, c) TOC and d) metabolomics of treated soil to check for growth inducing metabolites, as soil health indicators. It would also be great to check for metabolic pathways/predictions of the community (may be from community data using PicRust/FAPROTAX) or study the diversity of some functional genes (nifH, narG, nrfA, napA, phoB/C, etc) to iconclude on role of specific microbes/their metabolic groups.
Reviewer 2 Report
Comments and Suggestions for Authors
Dear Authors,
Your manuscript seems interesting because it show the manuscript presents the effect of ryegrass cultivation on increasing yields and fruit quality of sweet persimmon "Jiro". The layout of the articles became standard for research papers. However, I have a few comments, doubts and observations:
Detailed comments
Title: The title is more of a conclusion. It does not present the purpose of the research. The title of the manuscript should be edited.
Abstract: Explanations should be added to abbreviations that appear for the first time (L. 18). The abstract does not contain the purpose of the research. Complete it.
Introduction: This chapter is written exhaustively. However, there is a lack of clearly presented research purpose. The purpose of the research should be in a separate paragraph. Lines 75-86 should be reworded.
Results and Analysis. In my opinion, the title of the chapter should be changed to: "2. Results".
L. 102. SOM is not a total soil organism. I'm afraid this is a translator's error. Please correct the entire manuscript, including chapter 4. If this is not a translator's error, please detail what you mean by "total soil organism".
Table 1. Add unit to AP
Table 2. correct the table. It is illegible. "fungal" should be capitalized.
L. 124. “Chao1indexe“ insert a space.
Table 3. correct the table. It is illegible.
L. 179-181. Insert spaces.
Figure 4. Match the explanations with Figure 4. You should standardize the font (L. 194).
Discussion. L. 236. delete a space.
L. 310. use superscript and subscript.
Materials and Methods. Provide the name, city and country of the manufacturer of the devices used.
L. 352. In what solution was the soil pH determined? water? KCl? other solution?
Conclusions. Does the research have a practical dimension? Are farmers interested in this solution? How will farmers care for the turf? Will it be necessary to use plant protection products or on the contrary?
Good luck!
Sincerely yours
Reviewer
Author Response
Please see the attachmen

Round 2
Reviewer 2 Report
Comments and Suggestions for Authors
I accept in present form